# Recent Trends in the Production and Recovery of Bioplastics Using Polyhydroxyalkanoates Copolymers

**DOI:** 10.3390/microorganisms12112135

**Published:** 2024-10-24

**Authors:** Andrés García, Claudia Aguirre, Andrés Pérez, Sofía S. Bahamonde, Viviana Urtuvia, Alvaro Díaz-Barrera, Carlos Peña

**Affiliations:** 1Centro de Investigación en Biotecnología, Universidad Autónoma del Estado de Morelos, Cuernavaca 62210, Morelos, Mexico; andres.garcia@uaem.edu.mx; 2Departamento de Ingeniería Celular y Biocatálisis, Universidad Nacional Autónoma de México, Cuernavaca 62210, Morelos, Mexico; claudia.aguirre@ibt.unam.mx; 3Escuela de Ingeniería Bioquímica, Pontificia Universidad Católica de Valparaíso, Valparaíso 2340025, Chile; aaperezb17@gmail.com (A.P.); sofia.soto.b@mail.pucv.cl (S.S.B.); viviana.urtuvia.gatica@gmail.com (V.U.)

**Keywords:** polyhydroxyalkanoates (PHAs), copolymers, fermentation, bioplastics, purification

## Abstract

Polyhydroxyalkanoates (PHAs) are polyesters synthesized as a carbon and energy reserve material by a wide number of bacteria. These polymers are characterized by their thermoplastic properties similar to those of plastics derived from the petrochemical industry, such as polyethylene and polypropylene. PHAs are widely used in the medical field and have the potential to be used in other applications due to their biocompatibility and biodegradability. Among PHAs, P(3HB-*co*-3HV) copolymers are thermo-elastomeric polyesters that are typically soft and flexible with low to no crystallinity, which can expand the range of applications of these bioplastics. Several bacterial species, such as *Cupriavidus necator*, *Azotobacter vinelandii*, *Halomonas* sp. and *Bacillus megaterium*, have been successfully used for P(3HB-*co*-3HV) production, both in batch and fed-batch cultures using different low-cost substrates, such as vegetable and fruit waste. Nevertheless, in recent years, several fermentation strategies using other microbial models, such as methanotrophic bacterial strains as well as halophilic bacteria, have been developed in order to improve PHA production in cultivation conditions that are easily implemented on a large scale. This review aims to summarize the recent trends in the production and recovery of PHA copolymers by fermentation, including different cultivation modalities, low-cost raw materials, as well as downstream strategies that have recently been developed with the purpose of producing copolymers, such as P(3HB-*co*-3HV), with suitable mechanical properties for applications in the biomedical field.

## 1. Copolymers: Short Chain Length and Medium Chain Length

Polyhydroxyalkanoates (PHAs) are long polymeric molecules that contain ester linkages in their structure and are widely used in the medical field. They have the potential to be used in other applications due to their biocompatibility and biodegradability. The PHA family can be classified structurally into two groups: short-chain-length (scl) (three to five carbons) and medium-chain-length (mcl) (six to twelve carbons) groups. In addition, PHAs can exist as homopolymers or copolymers, depending on the similarity or diversity of their monomer unit composition (Figure 1). This is determined by the specificity of the PHA synthase enzymes involved in their synthesis [1]. Examples of short-chain-length PHAs include poly(3-hydroxybutyrate) P(3HB), poly(4-hydroxybutyrate) P(4HB), and poly(3-hydroxyvalerate) P(3HV) or the copolymer P(3HB-*co*-3HV). In contrast, examples of medium-chain-length PHAs include homopolymers such as poly(3-hydroxyhexanoate) P(3HHx), poly(3-hydroxyoctanoate) P(3HO), poly(3-hydroxydodecanoate) P(3HDD) and copolymers, such as P(3HHx-*co*-3HO). P(3HB), a homopolymer comprising four carbon subunits of 3-hydroxybutyrate (3HB), is the most prevalent and extensively studied member of the PHA family [2,3,4].

On the other hand, the fermentation process allows for the production of PHAs that exhibit a wide range of structural variations, encompassing over 150 types of HAs. The affinity between the microorganism and the growth substrate, the specificity of the corresponding PHA synthase of each bacterial strain used and the carbon sources fed during the fermentation process have a significant impact on the length of the polymer chain of the produced PHA. These factors serve as guides for the synthesis of homopolymers, copolymers or blends [5]. Regarding the substrate specificity of the PHA synthase, which is responsible for the composition of the monomer, a limited number of PHA synthases with a broad substrate specificity have been identified in *Aeromonas caviae* and *Pseudomonas stutzeri*. These PHA synthases were then expressed in a PHA-negative mutant of *Ralstonia eutropha*, resulting in the production of scl and mcl PHA copolymers [6]. For example, combinatorial mutations in *Pseudomonas putida*, *Pseudomonas oleovorans* and *Pseudomonas aeruginosa* that enable the synthesis of PHAs with 3HB, 3HHx, 3HO, 3HD or 3HDD monomers have been described [7].

## 2. Thermal and Mechanical Properties of PHA Copolymers

PHAs exhibit a wide range of physical and chemical properties, which are influenced by monomer composition and chain length. Monomer composition has a significant effect on the hydrophobicity, melting point temperature (T_m_), glass transition temperature (T_g_) and degree of crystallinity of the polymer [8,9]. Table 1 compiles the thermal and mechanical properties of PHA copolymers. For example, mcl PHAs are thermo-elastomeric polyesters that are typically soft and flexible with low to no crystallinity. In contrast, scl PHAs are highly crystalline, hard and brittle; these properties limit their range of applications and processability. In comparison to scl PHAs, mcl PHAs exhibit several distinctive properties. These include a relatively low T_g_ and T_m_, a low tensile strength and modulus, and a higher elongation at break. Furthermore, the molecular weights of mcl PHAs are relatively lower than those of scl PHAs, with a range of 40,000 to 412,000 Da [10]. The polydispersity index (PDI) of mcl PHA copolymers ranges from 1.6 to 4.4. In addition, the PDI values for mcl PHAs with unsaturated monomers are comparatively higher than those with saturated monomers. One of the most important parameters for characterizing an mcl PHA is its T_m_. The melting peaks can be attributed to the melting of the crystalline regions present in the polymer. In contrast, the T_g_ describes the temperature at which a transition occurs from a hard, brittle glassy state to a viscous, rubbery state [11,12]. Mcl PHAs have T_g_ values ranging between −25 and 65 °C and possess a low T_m_, ranging from 42 to 65 °C. This combination of T_g_ values below room temperature and a low degree of crystallinity imparts elastomeric behavior to these polymers (Table 1). 

Mcl PHAs are the only biopolymers produced by microorganisms that exhibit thermoplastic elastomer properties and resemble natural rubber produced by *H. brasiliensis* [23].

Moreover, the properties of scl PHAs vary considerably depending on the composition of the monomer, the side chain length and the molar mass of these polymers. For example, P(3HB), with a T_m_ of 180 °C and a T_g_ of 5 °C, is highly crystalline, brittle and stiff, with a tensile strength comparable to that of polypropylene, whereas P(4HB), with a T_m_ of 54 °C and a T_g_ of −49 °C, is a malleable thermoplastic material with a tensile strength comparable to that of polyethylene [24,25]. In addition, the influence of diverse 4HB fractions within the P(3HB-*co*-4HB) matrix on the T_m_, T_g_ and storage modulus has been investigated, revealing a consistent reduction in these parameters as the 4HB content increases. While the yield stress and breaking stress exhibit only slight decreases when the 4HB fraction increases, the elongation at break demonstrates a notable increase, rendering these materials highly flexible and elastic. The characteristics of P(4HB) differ significantly from those of other scl PHA polymers, including P(3HB) and P(3HB-*co*-3HV). P(4HB) exhibits an exceptional degree of extensibility, with elongation at break values up to 100%, coupled with a high flexibility [26]. These properties render P(4HB) a promising candidate for surgical applications. On the other hand, the introduction of a monomer into the polymer backbone, as in the case of heteropolymers, greatly affects the polymer’s properties by increasing its flexibility and toughness and decreasing its stiffness. To illustrate, a copolymer such as P(3HB-*co*-3HHx) has a lower T_m_ and level of crystallinity and is more malleable than P(3HB). The reduction in these values in the copolymer is advantageous in providing an elastic and flexible texture at normal temperatures. It should be noted that these properties can be influenced by a number of factors, including the type of bacteria, the substrate, feeding strategies and physiological conditions. All of these components can be manipulated in order to attain a polymer with the desired properties.

On the other hand, it is crucial to consider three fundamental mechanical properties when evaluating the suitability of a polymer for a specific commodity application. For example, other polymers, such as poly(acrylic acid), are highly versatile due to their properties, such as their high hydrophilicity and superabsorbent capacity. This renders them an excellent choice for the development of bioactive and biocompatible hydrogels for biomedical applications [27]. With regard to PHA copolymers, the elongation at failure is a measurement of toughness and reflects the total deformation that the polymer can withstand before fracturing [28]. The Young’s modulus is a measure of stiffness that considers the slope of the stress response to deformation at very low strains. For example, a steeper slope at low deformation values is indicative of a stiffer material. Finally, the ultimate tensile strength is a measure of the maximum strength of the material prior to the onset of plastic deformation. This value is derived from the maximum of the initial peak in the stress versus strain diagram [29,30].

P(3HB) is insoluble in water and exhibits relative resistance to hydrolytic degradation. It exhibits low oxygen permeability and favorable thermoplastic properties, although its mechanical properties, including its Young’s modulus and tensile strength, are inferior to those of petroleum-based polymers such as polypropylene. The densities of crystalline and amorphous P(3HB) are 1.26 and 1.18 g cm^−3^, respectively, and the molecular weight of P(3HB) produced by wild-type bacteria is typically within the range of 10,000 to 3,000,000 Da [31]. P(4HB) is a robust and malleable thermoplastic material with a tensile strength comparable to that of polyethylene. It displays an elongation at a break of 100%; therefore, it exhibits highly elastic properties. The material properties of P(4HB) can be modified when combined with other hydroxy acids. This is demonstrated by the differing properties of P(3HB), P(3HB-*co*-3HV) and P(4HB). On the other hand, the incorporation of secondary different HA monomers other than 3HB into the polymer chain greatly improves the material properties of P(3HB), including its crystallinity, melting point, stiffness and toughness. The most well-known copolymer, P(3HB-*co*-3HV), exhibits several notable differences compared to P(3HB). These include a reduction in crystallinity, a decrease in the T_m_, an increase in the elongation to break, a decrease in the Young’s modulus and an increase in impact strength. The mole percentage of 3HV is a crucial factor in determining the properties of the copolymer. The thermomechanical properties of P(3HB-*co*-3HV) can vary considerably due to the wide range of compositions that can be achieved, from 0 to 30 mol% 3HV [32,33,34,35]. In comparison to P(3HB-*co*-3HV), the three-component polymers detailed in Table 1 have been observed to exhibit an increase in thermal stability. This is evidenced by an increase in the T_m_ and T_d_, as well as an enhancement in the melting point and glass transition temperatures, alongside a notable reduction in crystallinity. It can therefore be concluded that these PHA copolymers represent a promising material for drug delivery and for the production of functional, low-crystalline scaffolds for cell and tissue engineering technologies, among other potential applications [13,15,18,22].

## 3. Production by Fermentation of P(3HB-*co*-3HV) PHA Copolymers 

The production of PHAs dates back to the 1980s, using heterotrophic bacteria as the first organisms for production, such as *Bacillus*, *Cupriavidus*, *Pseudomonas* and *Azotobacter*, as well as using feedstock like corn or sugar cane, which has to be cultivated directly for bioplastic production (first-generation bioplastic) and causes agriculture-related environmental impacts (e.g., land use and competition between feedstock and food). In the second stage or generation, and with the purpose of reducing production costs, specifically the cost of substrates, new raw materials not suitable for food or feed, e.g., wood cellulose and waste materials from initial biomass processing, e.g., food waste, were incorporated.

More recently, studies have focused on the use of the byproducts generated during fermentation processes, such as glycerol, methane and carbon dioxide (CO_2_), as well as wastewater, using methanotrophic, halophilic organisms and also those that are capable of reducing CO_2_ in formic acid. On the other hand, photoautotrophic cyanobacteria using CO_2_ (e.g., from the air) as carbon and (sun-)light as energy are potential sources for the synthesis of PHAs. This last generation method could offer new opportunities to reduce environmental impacts and take advantage of the products generated in different fermentation processes in a circular and sustainable economy scheme, aimed at lowering production costs and making bioplastic production processes more environmentally friendly. Figure 2 describes the characteristics of each stage or generation, as well as the advantages and challenges that have been faced in each of these, highlighting the fact that in each generation, progress is made in obtaining a product at a lower cost and with a lower environmental impact. In the following sections, some examples related to microbial sources and raw materials, described in each generation, will be discussed.

### 3.1. Comparison of Bacterial P(3HB-co-3HV) Production

A schematic representation of P(3HB-*co*-3HV) production using different raw materials is shown in Figure 3. Studies in different bacteria have reported the production of P(3HB-*co*-3HV) in shake-flask, batch, extended batch and fed-batch cultures [20,36,37,38,39].

Different studies have attempted to establish the culture conditions in a laboratory fermenter to produce copolymers with a varied content of 3HV. The collected data (Table 2) highlight that in shake flasks, depending on the growth conditions, bacterium and the presence or absence of a co-substrate, it is possible to obtain P(3HB-*co*-3HV) with different 3HV monomer contents. For example, it is well established that in the case of P(3HB-*co*-3HV) produced by *Salinivibrio* sp. TGB10 (a moderately halophilic bacterium), increasing the propionate (C3) concentration decreases both the cell dry weight (CDW) and the P(3HB-*co*-3HV) content, while the 3HV fraction increases significantly, reaching up to 72.0 mol% when 8 g L^−1^ of C3 is added. This demonstrates the strong influence of the co-substrate on the polymer composition [38]. Similarly, in *Halomonas* strains, the use of co-substrates, as starch and propionate, in shake flasks has resulted in a 3HV fraction as low as 0.7 mol% with 1 g L^−1^ of propionate, but increasing the concentration to 5 g L^−1^ can raise the 3HV fraction to 5.4 mol% [37]. In contrast, batch fermenter experiments with *A. vinelandii* OP show a 3HV content up to 35 mol% when using sucrose and valeric acid (C5) as carbon sources [40]. This is consistent with findings in shake flask experiments, in which the addition of valeric acid also promoted higher 3HV fractions, although the overall polymer content was lower. Moreover, extended batch fermenter experiments have shown that adjusting aeration and agitation rates can modulate the 3HV fraction from 9.1 to 20.8 mol%. Nevertheless, the synthesis of P(3HB-*co*-3HV) with variations in 3HV content not only depends on the microorganism and carbon sources but also on the specific fermentation conditions, such as the modality (batch, fed-batch or extended batch) and operational parameters like agitation and aeration rates. These parameters can lead to the generation of P(3HB-*co*-3HV) structures with different mechanical and physical properties, which expand the range of applications. It has been described that P(3HB-*co*-3HV) copolymers with >20 mol% of 3HV are most suitable for industrial applications, such as nanofibrous scaffolds and drug delivery carriers [41,42]. Regarding the mode of operation, differences are observed in the content and composition of the polymer. In fed-batch systems, Tao et al. [38] achieved more robust yields; for instance, *Salinivibrio* sp. TGB10 produced up to 81.8% w w^−1^ of P(3HB-*co*-3HV) with a 3HV fraction of 22.6 mol% when grown in the presence of 50 g L^−1^ of propionate.

Furthermore, *B. megaterium* NRRL B-14308 (recombinant) produces up to 80% w w^−1^ of P(3HB-*co*-3HV) with a high 3HV fraction (58 mol%), indicating the potential of this strain in high-density fed-batch cultures [43]. These results highlight that fed-batch systems, due to controlled feeding and optimized conditions, can significantly improve both the polymer content and the monomer composition. It is interesting to point out that studies on P(3HB-*co*-3HV) production have only been performed on a laboratory scale. An important challenge is to develop processes for production on a greater scale. In this way, a major obstacle in P(3HB-*co*-3HV) production on an industrial level, if you want to scale up production, consists of the high cost of carbon sources. One potential solution to this problem is the production of P(3HB-*co*-3HV) through the use of residual carbon sources.
microorganisms-12-02135-t002_Table 2Table 2Comparison of P(3HB-*co*-3HV) production using different microorganisms and strategies of cultivation.ModeMicroorganismCarbon Source/*co*-Substrate Concentration (g L^−1^)Operational ConditionsCDW (g L^−1^)P(3HB-*co*-3HV) (g L^−1^)P(3HB-*co*-3HV) (% w w^−1^)3HV Fraction (mol%)Ref.Shake flasks (batch)*Halomonas alkalicola*Glu150 rpm3.181.445.37.7[36]*Halomonas bluephagenesis*TD01/p341-amy03713-glu04552Starch + C3 (1)200 rpm10.25.251.00.7[37]Starch + C3 (5)200 rpm9.14.953.75.4*Halomonas. bluephagenesis* TY194Glu+ Gluconate200 rpm6.34.165.025.0[44]*Salinivibrio* sp. TGB10Glu (20) + C3 (2)200 rpm, 30 °C8.04.150.927.3[38]Glu (20) + C3 (4)200 rpm, 30 °C7.13.245.348.4Glu (20) + C3 (6)200 rpm, 30 °C5.72.340.865.1Glu (20) + C3 (8)200 rpm, 30 °C4.91.529.772.0*Azotobacter vinelandii* OPSuc (20) + C5 (1)200 rpm, 30 °C4.52.860.927.4[40]Suc (20) + C7 (1)200 rpm, 30 °C3.31.340.822.3*Salinivibrio* sp. TGB11Glu (20) + C3 (1.5)200 rpm, 30 °C6.43.14927.4[45]Batch fermenter*Cupriavidus necator* H16Fru (20) + C5DOT 30%, 0.83 vvm14.65.336.313.3[46]*Azotobacter vinelandii* OPSuc (20) + C5 (1)600 rpm, 1 vvm5.23.363.835[40]Suc (20) + C5 (1)300 rpm, 1 vvm5.03.873.318.6*Azotobacter vinelnadii* OPNASuc (20) + C5 (1)300 rpm, 1 vvm1.31.1856.6[39]Suc (20) + C5 (1)700 rpm, 1 vvm2.01.6808.3Suc (20) + C5 (4)700 rpm, 1 vvm1.9

21Fed-batch fermenter*Salinivibrio* sp. TGB11Glu (20) + C3 (1.5)500 rpm, 1 vvmfeed: Glu(200) + C3 (300)100.359.359.13[45]*Salinivibrio* sp. TGB10Glu (500) + C3 (50)400–800 rpm, 1 vvm33.527.481.822.6[38]*Halomonas mediterranei* DSM 1411Glu + C4:C5 (56:44%mol)200–800 rpm,0.75 vvm5.92.237.738.3[14]
Glu + C4:C5 (56:44%mol) + Tween 80^®^200–800 rpm,0.75 vvm6.84.058.942.6
*Bacillus megaterium* NRRL B-14308 (recombinant)Glu (20)exponential feeding (Glu: 100)300 rpm (variable)7.76.180.058.0[43]Extended batch fermenter*Azotobacter vinelandii* OPSac (20) + C5 (1)450 rpm, 0.3 vvm2.71.038.69.1[20]Sac (20) + C5 (1)450 rpm, 1.0 vvm6.54.061.720.8Sac (20) + C5 (1)450 rpm, 2.1 vvm6.84.871.19.1Suc: sucrose, Glu: glucose, Fru: fructose, C3: propionate, C4: butyrate, C5: valerate, C7: heptanoate.


### 3.2. P(3HB-co-3HV) Production from Waste Carbon Sources

As mentioned, plastic products are derived from fossil fuels, leading to severe environmental concerns [47]. P(3HB-*co*-3HV) production using waste carbon from agro-industrial sources presents a great alternative to minimize the production costs of bioplastics; however, its use depends on prior treatment. The process can consist of washing, drying, rupturing, spraying, sifting, hydrolysis, filtration and sterilization. Figure 4 shows the main pretreatments of different agricultural wastes according to the kind of waste. To incorporate agricultural waste (crop waste, animal waste, processing waste and hazardous waste), it is necessary to carry out some pretreatments to integrate the different nutrients into the broth culture. All waste has different structures and compositions; therefore, treatments must be designed that take into account the nature of the waste. 

The data collected from the past five years (Table 3) highlight that major studies have been reported using shake flasks. For example, Tian et al. [48] showed P(3HB-*co*-3HV) production by *Photobacterium* sp. TLY01 up to 4.2 g L^−1^ in shake flasks, with a 3HV fraction up to 79 mol% when using soybean oil or plant oil and valerate as substrates. Likewise, these authors reported P(3HB-*co*-3HV) production in fed-batch cultures, showing that the strain produces up to 16.3 g L^−1^ (CDW = 19.4 g L^−1^) with a 3HV fraction of up to 69 mol%. These authors observed that it is possible to synthesize P(3HB-*co*-3HV) using low-cost substrates from plant oils, such as soybean and corn starch, which could be a promising option for industrial applications. 

Nygaard et al. [49] reported that P(3HB-*co*-3HV) production using (untreated) residual glycerol (a byproduct generated by the biodiesel industry) and *C*. *necator* ATCC17697 was able to produce P(3HB-*co*-3HV) up to 3.2 g L^−1^ under optimal substrate concentration conditions, with a 3HV fraction up to 7.6 mol% for shake-flask fermentation. In another study, Zhao et al. [50] reported P(3HB-*co*-3HV) production using three carbon substrate combinations, with the volatile fatty acid (VFA) + C5 (10 g L^−1^) mixture achieving the highest 3HV fraction (83.4 mol%) and a P(3HB-*co*-3HV) production of 2.48 g L^−1^ using *Paracoccus* sp. TOH.
microorganisms-12-02135-t003_Table 3Table 3Comparison of P(3HB-*co*-3HV) production using different microorganisms and waste carbon sources.ModeMicroorganismCarbon Source/*co*-Substrate Concentration (g L^−1^)Operational ConditionsCDW(g L^−1^)P(3HB-*co*-3HV)(g L^−1^)3HV Fraction (mol%)Ref.Shake flasks *Paracoccus* sp. TOHGly (5) + C3 (3)200 rpm3.081.587.3[50]Gly (5) + C5 (3)200 rpm3.762.4846.8*Bacillus megaterium*PPH (18) + C5 (2)200 rpm3.41.76[51]PPH (10) + C5 (10)200 rpm2.40.720*Cupriavidus necator* ATCC 17697Gly (20) + C3 (1)150 rpm4.93.27.6[49]Gly (20) + C3 (2)150 rpm4.82.99.2Gly (20) + C3 (4)150 rpm3.82.011.8*Vibrio alginilyticus* LHF01Gly (20) + C3 (4)200 rpm8.61.424.4[52]*Photobacterium* sp. TLY01
Soybean oil (20) + C3 (8)200 rpm7.53.032.9[48]Soybean oil (20) + C5 (8)200 rpm8.64.279.0Corn starch (10) + C5 (8)200 rpm10.83.244.8*Cupriavidus necator* DSM 545Melon (50)145× *g*5.92.016.9[53]Batch fermenter*Cupriavidus necator* H16Glu + cassava peel200–500 rpm, 1 vvm3.40.9735.5[54]Fed-batch fermenter*Photobacterium* sp. TLY01Soybean oil (20) + C5 (5)500 rpm, 1 vvm19.4116.369.1[48]Gly: waste glycerol, PPH: pineapple peel hydrolyzed, Glu: glucose, LA: levulinic acid, C3: propionate, C5: valerate, VFA: volatile fatty acid.


### 3.3. P(3HB-co-3HV) Production with Biogas as Carbon Source

Several *Methanotrophic* bacterial strains, such as *Methylocystis*, *Methylosinus*, *Methylocapsa*, *Methylobacterium* and others, are capable of producing PHAs. Several strains, such as *Methylocystis* and *Methylosinus*, have demonstrated the ability to metabolize methane (CH_4_) into P(3HB-*co*-3HV) using propionic acid (C3) and/or valeric acid (C5) as co-substrates (Table 4). The production of P(3HB-*co*-3HV) from CH_4_ is an emerging and highly promising area of research due to the abundance and low cost of CH_4_, as well as its potential to reduce greenhouse gas emissions [55,56]. Moreover, this substrate enables working in non-sterile conditions, which contributes to reducing the operating costs of the process [57]. Recent studies have employed various cultivation methods, including batch cultures in bottles and bioreactors, and semi-continuous bioreactors. Gęsicka et al. [58] reported that the concentration of CH_4_ significantly influences biomass production, while the 3HV fraction remained relatively constant regardless of the microbial consortium used. On the other hand, the pure bacterium *Methylocystis hirsuta* DSM 18500 exhibited an extremely high 3HV fraction (88 mol%), highlighting its potential for synthesizing highly specialized copolymers. Recently, in addition to CH_4_ as a carbon source, the use of ethane (C_2_H_6_) and/or CO_2_ has been explored. Myung et al. [15] used *M. parvus* OBBP to produce P(3HB-*co*-3HV) with a 3HV fraction of up to 25 mol%, using a gas mixture of C_2_H_6_:O_2_ (at a ratio of one to four). This finding is noteworthy, as it indicates that ethane could be utilized in the synthesis of P(3HB-*co*-3HV). Tarawat et al. [59] used *Nostoc microscopicum* to produce P(3HB-*co*-3HV) using different CO_2_:acetate ratios as carbon sources, reaching up to 11 mol% of 3HV; and using different CO_2_:C5 or C3 ratios can produce P(3HB-*co*-3HV) with higher fractions of 3HV (between 31 and 96 mol%).

Regarding batch fermentation, studies have found that the CH_4_:C5 ratio and C5 addition frequency are key factors in determining the 3HV fraction. Tran et al. [60] observed a significant increase in the 3HV fraction with increased C5 addition frequency. On the other hand, Lee et al. [61] highlighted the impact of pH levels and C5 concentration on P(3HB-*co*-3HV) production and composition, finding that a pH of 6.5 is optimal for maximal production, though a lower pH favors a high 3HV fraction (22.4 mol%). The use of biogas (CH_4_ or C_2_H_6_) as a carbon source for P(3HB-*co*-3HV) production offers a sustainable and economically viable approach to biopolymer production. Cultivation strategies such as pH modification and the manipulation of CH_4_ and C5 concentrations are crucial for optimizing production and copolymer composition. Future projections suggest that optimizing these parameters, along with developing more efficient microbial strains, will enable the large-scale production of P(3HB-*co*-3HV) with specific properties, aligning with sustainability and circular economy goals. An interesting approach to improving existing applications using gases as a carbon source is to enhance their availability in the culture medium using gas solubilization methods, such as the use of nanobubbles for methane supply [62].

## 4. Extraction and Purification of PHAs 

It is important to point out that the primary challenge in the commercial production of PHAs lies in the polymer separation stages, which remain complex and inefficient, leading to high production costs [63]. The development of a bioprocess that allows for the recovery of PHAs through a simple, efficient and minimally polluting strategy is essential to positively impact the viability of the commercial production of the biopolymer. Furthermore, the designed extraction process must provide a high performance and have low costs because this stage can represent up to 35% of the total cost of the product [7]. 

PHAs are intracellular products; therefore, in general, three stages are required to obtain the final product: (1) cell breakdown; (2) the extraction or separation of PHA granules from cellular remains; and (3) the purification of the biopolymer granules. In general, three methods of cell breakdown can be identified: (1) the mechanical disruption of the residual biomass; (2) the chemical digestion of residual biomass by the application of chemical products; and (3) biological digestion by adding biocatalysts or acellular components that can lyse cells such as bacteriophages [64,65,66] (Figure 5). 

Among mechanical methods for cell disruption, bead mills, high-pressure homogenizers and ultrasonication are the most prevalent at the industrial scale. These methods are favored mainly because of the minimal damage they cause to products and the environment. However, several disadvantages must be considered, including their high capital investment cost, lengthy processing time and difficulty in scaling. High-pressure homogenization is a widely used method for PHA extraction [67]. This method involves forcing liquid through a narrow nozzle at a high pressure of 15 to 40 bars and shear stress. Several forces act on it, like cavitation, a high shear impact and turbulence, resulting in the homogenization of the sample. Although this method offers a high efficiency in breaking the cells, both the investment and operation costs, especially for larger-scale equipment, impact the economy of the process and the final production cost. Another method used is ultrasound waves. This process involved the treatment of the cells with an ultrasonic sonicator to create a suspension of polymer granules. This suspension is then freeze-dried and pulverized, before subjecting it to an air classifier. When this sample was introduced into the classifier, it produced a fine fraction of 38% and a coarse fraction of 62%. The fine fraction was then subjected to chloroform extraction along with methanol precipitation to obtain P(3HB) particles [67]. This method is favored mainly because of the low level of damage it causes to the polymer and to the environment; however, like the mechanical rupture method, its high investment costs and difficulty in scaling make it unviable on a commercial scale.

The dissolution of the residual biomass and the intact conservation of the PHA granules are the primary objectives of the chemical digestion method, mainly through the application of chemical products, such as treatments with acids or bases [68]. In previous studies using this kind of method, strong oxidation agents such as hypochlorite and sodium hydroxide were employed to dissolve residual biomass [68]. However, if the concentration of the oxidizing agent is not well controlled, when dissolved, it not only disintegrates the residual biomass but also affectshe PHA, which leads to a low level of recovery and causes breakage in the chains, directly affecting the molecular weight of the polymer [69]. This non-selective method was later replaced by one in which the digestion of the residual biomass is selective, through a series of agents such as anionic surfactants (such as SDS) and acids [70]. However, although the biopolymer resulting from chemical digestion has a high level of purity, the environmental impact is also high due to the large amounts of solvents and surfactants required [71]. Furthermore, due to the high concentration of anionic surfactants used, the recovery step could be affected by high production costs [70]. Solvent extraction is a common method to recover PHAs since it has a high level of efficiency and allows for the elimination of endotoxins from the recovered biopolymer [72]. Currently, most PHA extraction strategies are based on halogenated solvents, such as chloroform, dichloroethane, chloropropane and methyl chloride, with chloroform being the most used. However, the use of these solvents has disadvantages since they are expensive, harmful to the environment, cause the degradation of the biopolymer and require large amounts of solvents [69].

A more environmentally friendly alternative is to use non-halogenated solvents such as acetone and various alcohols. Bartels et al. [73] carried out a study focused on the evaluation of non-halogenated solvents, especially acetone for the recovery of poly(3-hydroxybutyrate-*co*-3-hydroxyhexanoate)—P(HB-*co*-HHx)—with an mcl HHx content >15 mol% and an MW average from 1.3 to 1.6 × 10^5^ Da. The results revealed that the use of acetone with a 2-propanol fraction of up to 30% was still able to extract the polymer 95% as efficiently as pure acetone. Additionally, when acetone and ethyl acetate were used in a multi-stage extraction process, a two-stage process was sufficient to extract 98–99% of the polymer from the cells. The authors claim that the developed strategy could help to decrease PHA production costs and contribute to a more ecological process, which would support the commercialization process of PHAs [73].

Enzymatic digestion is a gentle technique; therefore, it is a selective separation method that has attracted the interest of many researchers, including patents for which this type of development is protected. Several classes of hydrolytic enzymes, mainly proteases, have been used. It is an attractive method due to its moderate operating conditions and because the enzymes are highly specific to their substrate; therefore, it is possible to obtain a polymer with a high degree of purity without affecting its molecular weight. Nevertheless, the enzymatic digestion techniques utilized to extract PHAs are exceedingly costly, predominantly due to the necessity for exogenous protein supplementation and the incorporation of additional purification steps. On an industrial scale, the prevalent methodologies necessitate substantial enzyme inputs, which present a significant challenge in reducing recovery costs. Furthermore, the utilization of solvents is a concern from an environmental standpoint [64].

### 4.1. Lytic Systems for the Extraction of PHAs

The ability of microorganisms to express their lytic proteins and therefore self-lyse has several advantages: on the one hand, it represents a decrease in polymer recovery costs since it helps release the product to the environment without the use of any chemical/enzymatic agent. Therefore, washing steps are reduced to obtain a clean polymer, and the impact on the environment is reduced.

Lysis systems utilized by bacteriophages that infect Gram-negative bacteria have been proposed as an alternative method for the release and purification of PHAs. This is due to the high degree of substrate specificity exhibited by the proteins involved, which do not affect PHAs. The final stage of the bacteriophage infection process is the lysis of the host cell. Double-stranded DNA phages require the involvement of various proteins to achieve this, and the process is one that is carefully regulated and temporally programmed. In Gram-negative hosts, this lysis process is divided into three steps: the permeabilization of the inner membrane, degradation of the peptidoglycan and degradation of the outer membrane. Each of these steps is carried out by at least one different protein, and in some cases, up to five enzymes are involved [74].

Several proteins involved in cell lysis using bacteriophages have been studied to develop bacteria capable of releasing PHAs into media. Among these proteins, two of them have shown promising results: the enzymes holin and endolysin. The genes have been obtained from various bacteriophages, and self-destructive strains have been cloned into PHA-producing bacteria and therefore have facilitated the expulsion of the polymer to the extracellular medium. The percentage of product recovered depends on several factors, among them are the type of holin and endolysin used, the inducing molecule and the promoter chosen in each case, and the method to recover the polymer once the cells have been lysed [75]. Despite the potential of lytic systems for PHA extraction, most of them are in the experimental phase and there are very few examples of cases in which they have been used on a larger scale. It is clear that more research is required to support the viability of using these methods on an industrial scale.

### 4.2. Purification of PHAs

The most common methods for the purification of this biopolymer are aqueous treatments with hydrogen peroxide combined with chelating agents to eliminate impurities. However, these treatments have the following drawbacks: very high operating temperatures (80–180 °C), which require the intense heating and cooling of the product and, in some cases, high-pressure equipment; and peroxide instability in the presence of high levels of cellular biomass. Prolonged elevated temperatures coupled with hydrogen peroxide can cause a severe decrease in the molecular weight of PHAs and, in some cases, can promote crystallization, which is undesirable for the production of an amorphous polymeric latex [76,77]. On the other hand, the disadvantage of the use of this type of solvent for the PHA purification process is its high level of environmental impact, so we must look for new, more environmentally friendly alternatives that allow us to obtain bioplastics with a high degree of purity. An alternative are the methods in which ultracentrifugation systems and/or diafiltration processes are employed. For example, the use of ozone purification has been proposed to increase the purity level of PHAs. To do this, ozone is applied to the biomass or solution in an oxygen flow with two and five % ozone. This treatment has favorable effects in terms of the whitening, deodorization and solubilization of impurities and thus facilitates their elimination from aqueous suspensions [71,78].

## 5. Conclusions

In summary, this review describes several aspects of copolymer P(3HB-*co*-3HV) production and recovery, focusing on different microorganisms and fermentation strategies. Despite the growing interest and increasing commercialization of PHAs, there are still challenges to producing biopolymers with properties similar to those of conventional plastics. Copolymers such as P(3HB-*co*-3HV) and P(3HB-*co*-3HHx) exhibit similar or even superior properties to plastics obtained from petroleum, expanding the potential application of these polymers, especially in the medical field.

The development of cultures in bioreactors permits the production of P(3HB-*co*-3HV) with a defined 3HV content, which leads to the generation of biomaterials of different mechanical and physical properties. In the literature, the production of P(3HB-*co*-3HV) has only been described to the level of the laboratory scale, and accordingly, it is necessary to increase efforts to develop cultures on a bioreactor pilot scale.

In recent years, new microbial sources have been proposed, among them, extremophilic organisms, such as methanotrophic and halophilic organisms, stand out as they are capable of producing PHAs in non-sterile culture conditions and can be easily implemented on a large scale, thus positively affecting production costs. On the other hand, the incorporation of low-cost raw materials, including agro-industrial waste and gases such as methane and CO_2_, can enhance the economic viability and sustainability of large-scale operations, thereby enhancing their competitiveness in economic terms. Furthermore, the implementation of environmentally conscious separation techniques that facilitate solvent recycling during extraction could reduce PHA production costs and lead to a more ecological process, thereby facilitating the broader commercialization of PHAs.

## Figures and Tables

**Figure 1 microorganisms-12-02135-f001:**
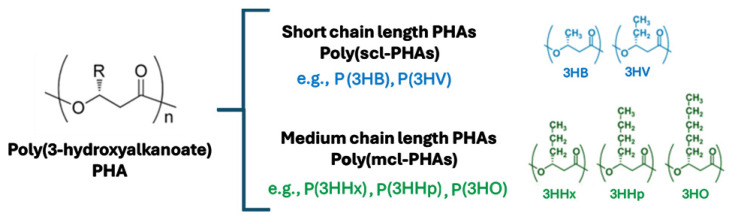
Representative examples of monomers’ PHA copolymers.

**Figure 2 microorganisms-12-02135-f002:**
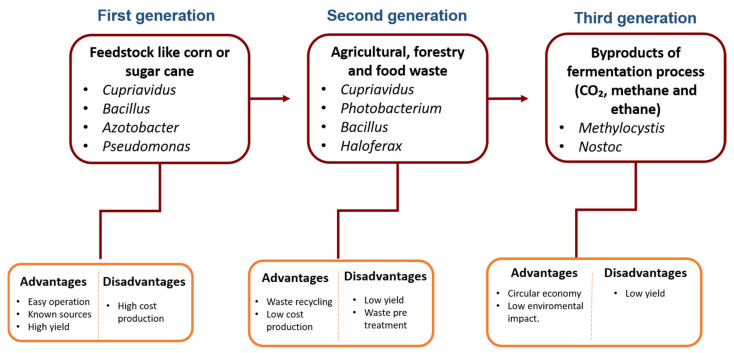
Overview of the different generations or stages in the production of PHAs by fermentation.

**Figure 3 microorganisms-12-02135-f003:**
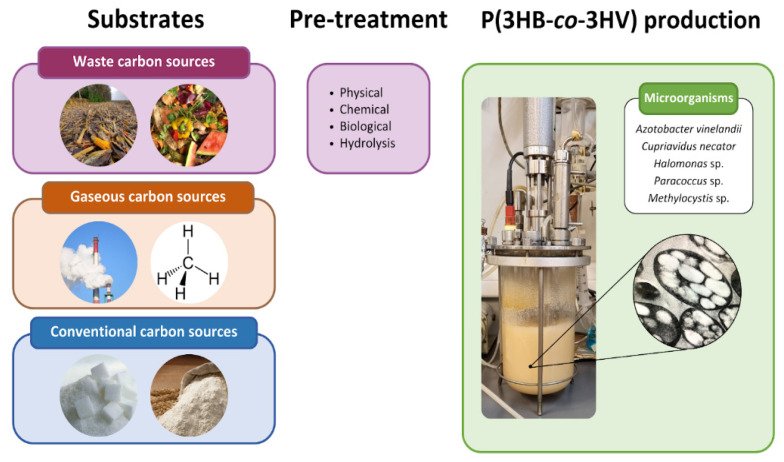
Schematic representation of P(3HB-*co*-3HV) production process using different carbon sources.

**Figure 4 microorganisms-12-02135-f004:**
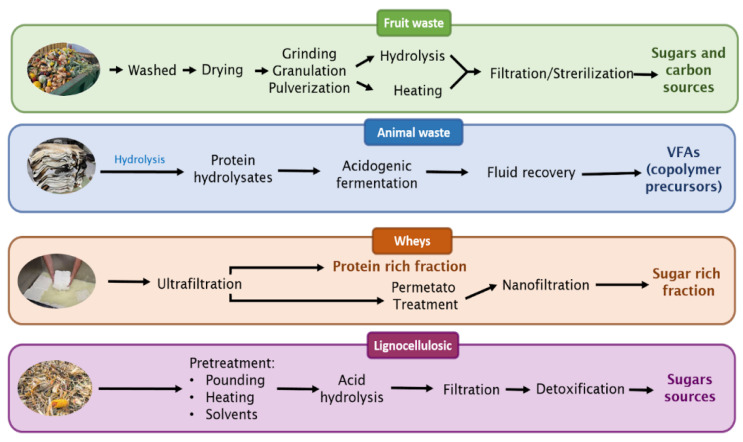
General scheme for the treatment of agricultural waste to obtain carbon sources.

**Figure 5 microorganisms-12-02135-f005:**
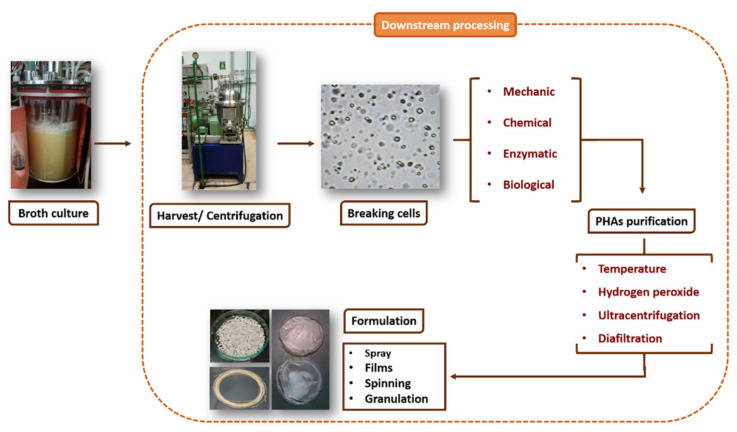
General scheme of the different PHA recovery methods.

**Table 1 microorganisms-12-02135-t001:** Thermal and mechanical properties of PHA copolymers.

Bacteria	PHA Composition (mol%)	Molecular Weight (kDa)	Thermal Properties				Mechanical Properties			Ref.
			T_m_ (°C)	T_g_ (°C)	T_c_ (°C)	T_d_ (°C)	Tensile Strength (MPa)	Young’s Modulus (MPa)	Elongation at Break (%)	
	^a^ P(3HB) (100)	900	179	3.1	41.2	285	18	1470	3.0	[13]
	^b^ P(3HV) (100)		106.2	−15.7	57.9	-	6.6	389	3.5	
*Haloferax mediterranei*	P(3HB/3HV) (74/26)	2820	108	-	56	266	15	782	7	[14]
*Methylocystis parvus OBBP*	P(3HB/3HV) (76/24)	132	147	−1	-	-	22	1000	50.5	[15]
	P(3HB/4HB) (97/3)	133	148	−2	-	-	35.6	1200	176	
	P(3HB/4HB) (90.5/9.5)	122	135	−5	-	-	31.2	800	284	
	P(3HB/6HHx/4HB) (97.6/1.4/1.0)	126	150	−1	-	-	27.6	700	134	
*Pseudomonas putida KTOY06ΔC*	P(3HB/HV/HHp) (71/13/16)	450	171	−24	51	-	8	366	63	[13]
	P(3HB/HV/HHp) (73/13/14)	370	-	−7.3	-	-	7	127	462	
*Pseudomonas entomophila*	P(HDD/H9D) (30/70)	131	46	−55	-	-	3	-	138	[16]
*Cupriavidus* sp. USMAA1020	P(3HB/4HB) (5/95)	156	62	−49	-	-	23	-	463	[17]
*Cupriavidus necator*	P(3HB/3HV/4HHx) (38/60/2)	250	81.87	−13	-	-	385	14	421	[18]
	P(3HB/3HV/4HHx) (71/26/3)	160	131	−5.6	-	-	141	12	324	
	P(3HB/3HV/4HHx) (80/16/4)	110	140	−3.5	-	-	541	20	321	
*Azotobacter chroococcum* 23	P(3HB/3HV) (90/10)	1531	163.2	−1.7	-	-	23.4	-	355	[19]
	P(3HB/3HV) (80/20)	1588	116.4	−6.3	-	-	21.5	-	690	
*Azotobacter vinelandii* OP	P(3HB/3HV) (81.4/18.6)	238	169.7	-	-	-	9.1	600	2.4	[20]
	P(3HB/3HV) (67.2/32.8)	434	166	0.64	52.8	-	22.6	1100	3.2	
	P(3HB/3HV) (63.3/36.7)	409	170	0.47	43.8	-	51.8	2200	5.1	
*Cupriavidus necator Re2160/pCB113*	P(3HB/3HHx) (68/32)	347	86	−1		278	8	101	856	[21]
	P(3HB/3HHx) (57/43)	117	88	−4		285	5	75	481	
	P(3HB/3HHx) (47/56)	120	-	−6		274	1	12	368	
	P(3HB/3HHx) (40/60)	211	-	−11		278	0.5	3	424	
	P(3HB/3HHx) (30/70)	227	-	−12		278	0.5	1	1075	
*Cupriavidus necator* B-10646	P(3HB/3HV/4HB) (55.2/18.5/26.3)	669	171	-	-	282	8.1	239.3	231.5	[22]
	P(3HB/3HV/4HB) (26.2/13.4/60.4)	507	158	-	-	274	10.1	37.5	371.1	
	P(3HB/3HV/3HHx) (71.4/26.1/2.5)	529	175	-	-	274	10.8	312.3	73.2	
	P(3HB/3HV/3HHx) (84.6/1.8/13.6)	924	172	-	-	270	9.4	257.5	390.5	

^a,b^ Commercial product. T_m_ melting temperature, T_g_ glass transition temperature, T_c_ crystallization temperature, T_d_ decomposition temperature.

**Table 4 microorganisms-12-02135-t004:** Comparison of P(3HB-*co*-3HV) production from gas.

Mode	Microorganism	Gas Composition	Co-Substrate (g L^−1^)	Operational Conditions	CDW (g L^−1^)	P(3HB-*co*-3HV) (% w w^−1^)	3HV Fraction (mol%)	Ref.
Bottle	Consortia from landfill biocover soil	CH_4_ (10%)	C5 (0.1)	130 rpm	0.45	26	35	[58]
	Consortia from peat bog	CH_4_ (10%)	C5 (0.1)	130 rpm	0.40	25	36	
	Consortia from peat bog	CH_4_ (20%)	C5 (0.1)	130 rpm	0.56	9.0	41	
	Consortia from peat bog	CH_4_ (25%)	C5 (0.1)	130 rpm	0.24	19	37	
	Consortia from activated sludge	CH_4_ (10%)	C5 (0.1)	130 rpm	0.45	27	39	
	*Methylocystis hirsuta* DSM 18500	CH_4_ (10%)	C5 (0.1)	130 rpm	0.49	9.0	88	
	*Methylocystis parvus* OBBP	C_2_H_6_:O_2_ (ratio 1:4)	C5 (1.0)	150 rpm, 30 °C	-	12.9	25	[15]
Batch Fermenter	*Methylocystis* sp. MJC1	CH_4_:Air (ratio 3:7)	C5 (^a^ 0.1)	0.2–0.25 vvm, 30 °C, pH 6.5	-	50	9.0	[60]
		CH_4_:Air (ratio 3:7)	C5 (^a^ 0.15)	0.2–0.25 vvm, 30 °C, pH 6.5	-	53	11.3	
		CH_4_:Air (ratio 3:7)	C5 (^b^ 0.1)	0.2–0.25 vvm, 30 °C; pH 6.5	-	43	29.7	
Batch Fermenter	*Methylocystis* sp. MJC1	CH_4_:Air (ratio 3:7)	C5 (0.1)	0.2–0.4 vvm, pH 6.0	12.8	47.8	18.8	[61]
		CH_4_:Air (ratio 3:7)	C5 (0.1)	0.2–0.4 vvm, pH 6.5	26.7	51	9.7	
		CH_4_:Air (ratio 3:7)	C5 (0.1)	0.2–0.4 vvm, pH 6.8	19.1	43.6	12	
		CH_4_:Air (ratio 3:7)	C5 (0.15)	0.2–0.4 vvm, pH 6.5	20.3	53.3	11.3	
		CH_4_:Air (ratio 3:7)	C5 (0.2)	0.2–0.4 vvm, pH 6.5	15.2	59.3	22.4	
Semi-continuous fermenter	*Methylosinus*-Consortia	CH_4_:CO_2_ (ratio 9:1)	C3	200 rpm, 30 °C	-	3.5	22.6	[57]
		CH_4_:CO_2_ (ratio 9:1)	C5	200 rpm, 30 °C	-	14.1	65	

Supplemented every ^a^ 6 h and/or ^b^ 3 h, C3: propionate, C5: valerate.

## Data Availability

The data presented in this study are available upon request from the corresponding author.

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
