# Peer review of "Recent Trends in the Production and Recovery of Bioplastics Using Polyhydroxyalkanoates Copolymers"

_microorganisms, 2024, doi:10.3390/microorganisms12112135_

Round 1
Reviewer 1 Report
Comments and Suggestions for Authors
This article reviews recent trends in the production and recovery of bioplastics based in polyhydroxyalkanoates copolymers. Although the topic is interesting in its scientific field, there are some issues that require the authors’ attention to improve the quality of this particular manuscript before further consideration for publication in a high-quality journal “Microorganisms”.
Specific comments:
1. This review covers many aspects of PHA production, recycling and application, but lacks a cohesive structure. Please improve.
2. The current manuscript mainly summarizes existing research work, but does not provide sufficient insights. The authors simply list various findings without discussing their implications and performing comparative analyses.
3. The authors use a large number of tables to present the data. However, it is necessary to add some representative charts to highlight the trends on this specific research topic.
4. Please avoid repeating the details about the same microbial strains and PHA properties, e.g.: Cupriavidus necator and Azotobacter vinelandii as PHA-producing bacteria since they have already been discussed multiple times in different sections (production methods and copolymer properties).
5. This manuscript should highlight cutting-edge research outcomes or emerging trends in PHA production (e.g., methanotrophs and extremophiles). Please also emphasize their differences from those obtained via traditional methods.
6. As stated by the authors, it is crucial to consider the three fundamental mechanical properties when evaluating the suitability of a polymer for a specific commodity application. Nevertheless, this important claim is not supported by any appropriate documentation. If possible, please consider the inclusion of the following relevant case study (DOI: 10.1039/c1jm14211a) in the reference list to strengthen manuscript quality and enrich article content.
Author Response
1. This review covers many aspects of PHA production, recycling and application, but lacks a cohesive structure. Please improve.
R: Thanks for the reviewer comments. In the new version we have reviewed the text and we believe that the writing have now been improved. Additionally, we have included a figure about the different generations or stages in the production of PHAs by fermentation (section 3, figure 2) to enhance the structure of the manuscript.
2. The current manuscript mainly summarizes existing research work, but does not provide sufficient insights. The authors simply list various findings without discussing their implications and performing comparative analyses.
R: In attention to this observation. In the new version, we have enriched the discussion throughout the manuscript (section 2, lines 108-115; lines 148-155; section 4, lines 342-344, 350-352, 359-363, 393-398; section 4.1, lines 405-411, 423-426; section 4.2, lines 436-440).
3. The authors use a large number of tables to present the data. However, it is necessary to add some representative charts to highlight the trends on this specific research topic.
R: We appreciate this observation. In the new version of the manuscript a new figure (section 3), describing the different generations or stages in the production of PHAs by fermentation has been included, as well as a text where each of these stages is discussed. Lines 158-181.
4. Please avoid repeating the details about the same microbial strains and PHA properties, e.g.: Cupriavidus necator and Azotobacter vinelandii as PHA-producing bacteria since they have already been discussed multiple times in different sections (production methods and copolymer properties).
R: Thank you for your comment. Regarding the emphasis on bacteria such as Azotobacter, other models like Halomonas and Bacillus have been added in section 3.1.
Tables 1, 2 and 3 provide a summary of the literature, including all reported information on thermal and mechanical properties, culture methods, operational conditions, and PHBV copolymer production (including CDW, accumulation, and 3HV fraction). Therefore, we believe it is necessary to keep Azotobacter vinelandii and Cupriavidus necator in the tables and text, despite being repeated, as they represent independent studies with different results that enrich the state of the art. Lines 205-208 and 226-230.
5. This manuscript should highlight cutting-edge research outcomes or emerging trends in PHA production (e.g., methanotrophs and extremophiles). Please also emphasize their differences from those obtained via traditional methods.
R: Thanks for the reviewer comments. In the new version of the manuscript (section 3.3) highlighting the PHA production using biogas e.g. in Methanotrophs and photoautotrophic organisms (cyanobacterium) has been included. Lines 273-279 and 287-294.
6. As stated by the authors, it is crucial to consider the three fundamental mechanical properties when evaluating the suitability of a polymer for a specific commodity application. Nevertheless, this important claim is not supported by any appropriate documentation. If possible, please consider the inclusion of the following relevant case study (DOI: 10.1039/c1jm14211a) in the reference list to strengthen manuscript quality and enrich article content.
R: We appreciate the reviewer's suggestion and have incorporated the reference (Lai et al. (2012) in the new version of the manuscript (section 2, lines 118-121). We agree that this will reinforce the manuscript and provide further context, thereby enhancing the article's overall quality.
Reviewer 2 Report
Comments and Suggestions for Authors
The work presented is highly interesting and offers clear insights into the topic. The figures are particularly well done, helping to illustrate key points effectively and making the content more accessible. The structure and flow of the review are coherent, and the writing strikes a good balance between technical detail and readability.
The article provides a comprehensive review of the production of various biodegradable plastics using bacteria, including different types of PHA (polyhydroxyalkanoates). This is a highly relevant and timely topic, given the increasing demand for biopolymers as a response to the growing scarcity of petroleum and the shift toward a society that prioritizes biodegradable products. The topic is of great interest, as the need for sustainable alternatives is only expected to grow in the coming years.
While the article does not present new data, it does an excellent job of summarizing existing knowledge on the subject. The figures are particularly effective in illustrating key points, and the structure is clear and well-organized, making the review both informative and easy to follow.
Overall, the review is well-executed and provides a solid contribution to the subject. It was a pleasure to read, and I believe it offers valuable insights for further exploration. This review is a valuable resource for anyone interested in the field of bioplastics, and I believe it contributes significantly to the ongoing discussion on sustainable materials.
Author Response
The work presented is highly interesting and offers clear insights into the topic. The figures are particularly well done, helping to illustrate key points effectively and making the content more accessible. The structure and flow of the review are coherent, and the writing strikes a good balance between technical detail and readability.
The article provides a comprehensive review of the production of various biodegradable plastics using bacteria, including different types of PHA (polyhydroxyalkanoates). This is a highly relevant and timely topic, given the increasing demand for biopolymers as a response to the growing scarcity of petroleum and the shift toward a society that prioritizes biodegradable products. The topic is of great interest, as the need for sustainable alternatives is only expected to grow in the coming years.
While the article does not present new data, it does an excellent job of summarizing existing knowledge on the subject. The figures are particularly effective in illustrating key points, and the structure is clear and well-organized, making the review both informative and easy to follow.
Overall, the review is well-executed and provides a solid contribution to the subject. It was a pleasure to read, and I believe it offers valuable insights for further exploration. This review is a valuable resource for anyone interested in the field of bioplastics, and I believe it contributes significantly to the ongoing discussion on sustainable materials.
R: We are grateful for your positive feedback on the review's quality and potential value for others in the field. Thank you for taking the time to provide such detailed and constructive input.
Round 2
Reviewer 1 Report
Comments and Suggestions for Authors
The revised version has adequately addressed most of the critiques raised by this reviewer and is now suitable for publication in "Microorganisms".